# Phalloidin and DNase I-bound F-actin pointed end structures reveal principles of filament stabilization and disassembly

Micaela Boiero Sanders [1,3], Wout Oosterheert [1,3], Oliver Hofnagel[1], Peter Bieling [2] ✉ & Stefan Raunser [1] ✉

Actin filament turnover involves subunits binding to and dissociating from the filament ends, with the pointed end being the primary site of filament disassembly. Several molecules modulate filament turnover, but the underlying mechanisms remain incompletely understood. Here, we present three cryo-EM structures of the F-actin pointed end in the presence and absence of phalloidin or DNase I. The two terminal subunits at the undecorated pointed end adopt a twisted conformation. Phalloidin can still bind and bridge these subunits, inducing a conformational shift to a flattened, F-actin-like state. This explains how phalloidin prevents depolymerization at the pointed end. Interestingly, two DNase I molecules simultaneously bind to the phalloidin-stabilized pointed end. In the absence of phalloidin, DNase I binding would disrupt the terminal actin subunit packing, resulting in filament disassembly. Our findings uncover molecular principles of pointed end regulation and provide structural insights into the kinetic asymmetry between the actin filament ends.

Actin is an abundant and highly conserved protein that controls the shape and movement of all eukaryotic cells[1,2]. The actin molecule comprises four subdomains (SD1 – SD4) with a high-affinity adenine-nucleotide binding site nestled within a central cleft between these subdomains[3]. Monomeric actin (G-actin) can polymerize into filaments (F-actin), which form a polar, double-stranded helix with a right-handed twist. During this process, SD1 and SD2 rotate about 12.4° relative to SD3 and SD4, generating a 'flattened' actin subunit arrangement in the filament[4–7]. F-actin exhibits two non-equivalent ends, known as the 'barbed' and 'pointed' ends[8–11]. At the barbed end, which exposes SD1 and SD3, F-actin growth is fast under polymerization-promoting conditions. However, in the absence of G-actin, subunits also rapidly dissociate from the barbed end. In contrast, both F-actin polymerization and disassembly are substantially slower at the SD2- and SD4-exposing pointed end of the filament[8–11]. The structural basis of this kinetic asymmetry has long remained unclear.

The dynamics of actin filaments are regulated by a multitude of actin-binding proteins (ABPs) that fine tune different aspects of the actin turnover cycle, such as nucleation, elongation, capping, severing, depolymerization and nucleotide exchange[12,13]. DNase I is a classic example of an ABP that triggers rapid F-actin depolymerization. Besides its well-established role of degrading DNA in processes such as apoptosis[14,15], DNase I potently depolymerizes actin filaments by a dual mechanism: First, it binds and sequesters G-actin with high affinity, hindering its polymerization[16–18]. Second, DNase I also interacts with the pointed end of actin filaments to increase the rate of depolymerization[16,17,19–22]. This dual actin sequestration/depolymerization function of DNase I has been proposed to contribute to actin clearance at sites of tissue damage and necrosis[23]. Despite extensive research on the G-actin-binding mode of DNase I[3,24], its interaction with the F-actin pointed end leading to depolymerization is not understood. Kinetic studies suggest that two DNase I molecules can simultaneously bind at the pointed end[22], but structural evidence is lacking due to the transient nature of the DNase I – pointed end complex. Consequently, the precise mechanism by which DNase I binds the F-actin pointed end to promote rapid filament disassembly remains unclear.

[1]Department of Structural Biochemistry, Max Planck Institute of Molecular Physiology, 44227 Dortmund, Germany. [2]Department of Systemic Cell Biology, Max Planck Institute of Molecular Physiology, 44227 Dortmund, Germany. [3]These authors contributed equally: Micaela Boiero Sanders, Wout Oosterheert. ✉e-mail: peter.bieling@mpi-dortmund.mpg.de; stefan.raunser@mpi-dortmund.mpg.de

The dynamics of actin filaments can also be altered by small molecule inhibitors and toxins. For instance, the well-studied toxin phalloidin from *Amanita phalloides* is a rigid bicyclic heptapeptide that represents a prototypical F-actin-stabilizing molecule[25,26]. Phalloidin stabilizes filaments by binding in a pocket formed by three adjacent actin subunits[27–29] and prevents subunit loss from both F-actin ends[30]. Notably, phalloidin-stabilized filaments cannot be depolymerized by DNase I either[16,17].

While actin subunits at the barbed end arrange in a canonical F-actin conformation that can be stabilized by phalloidin[31,32], recent structural work has uncovered that the two terminal actin subunits at the pointed end adopt a twisted, G-actin-like conformation in the absence of phalloidin[32,33]. However, whether phalloidin can still bind between these terminal G-actin-like subunits and whether such binding requires larger conformational rearrangements is presently unclear. The mechanism by which phalloidin prevents actin depolymerization from the pointed end, even in the presence of DNase I, therefore remains incompletely understood.

Here, we use cryo-electron microscopy (cryo-EM) to investigate how DNase I and phalloidin affect the arrangement of the F-actin pointed end. We show that phalloidin rearranges the pointed end by binding in between the two terminal actin subunits, resulting in a stabilized arrangement, which explains why the toxin potently inhibits actin depolymerization. Interestingly, phalloidin binding induces a flattened conformation of the subunits at the pointed end that allows two molecules of DNase I to bind to the D-loops of the penultimate and ultimate actin subunits in a flexible arrangement. In the absence of phalloidin, however, DNase I binding to the penultimate pointed end subunit would lead to the destabilization and displacement of the last actin subunit from the filament, explaining the mechanism of actin depolymerization by DNase I.

## Results

### Phalloidin stabilizes and flattens the F-actin pointed end

We first set out to investigate how phalloidin affects the filament pointed end arrangement to prevent subunit dissociation. Making use of recent advances in our sample preparation protocols (see Methods)[31,34,35], we determined the cryo-EM structure of the undecorated pointed end of α-actin filaments at a resolution of 3.1 Å, as well as the pointed end of β-actin filaments in the presence of phalloidin at a resolution of 3.7 Å (Fig. 1a–c, Supplementary Figs. 1, 2, 3, Supplementary Movie 1, Supplementary Table 1).

The pointed end structures with and without phalloidin display the classical double-stranded arrangement of the actin filament. In both structures, we observed a flattened F-actin conformation for all internal subunits ($P_2$ and above) that engage in all available intersubunit interactions. In contrast to internal subunits, we found that the ultimate ($P_0$) and penultimate ($P_1$) subunits are in a twisted, G-actin-like conformation in the absence of phalloidin, confirming findings of a recent structural study[32] (Fig. 1d, Supplementary Fig. 4). In the phalloidin-bound structure, we observed clear density for phalloidin in all of its canonical binding sites constituted by three consecutive

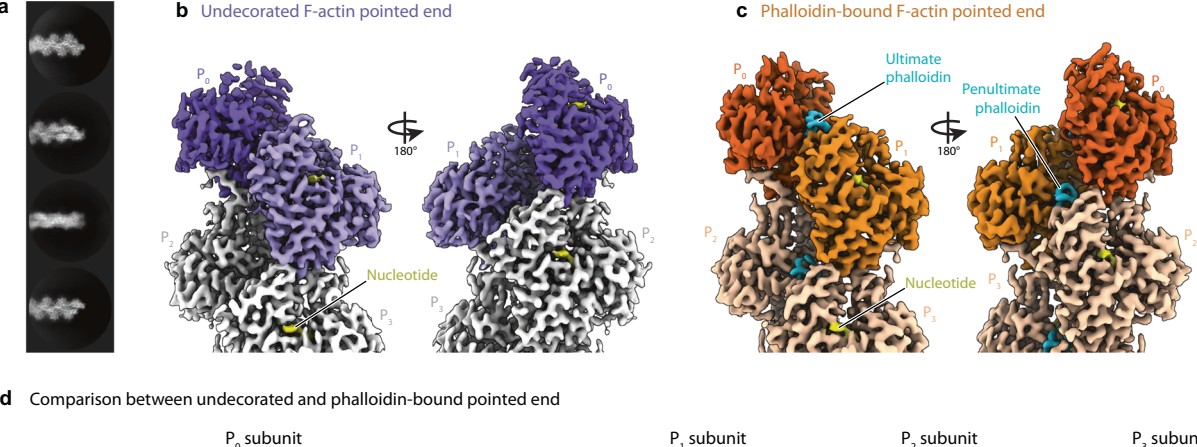

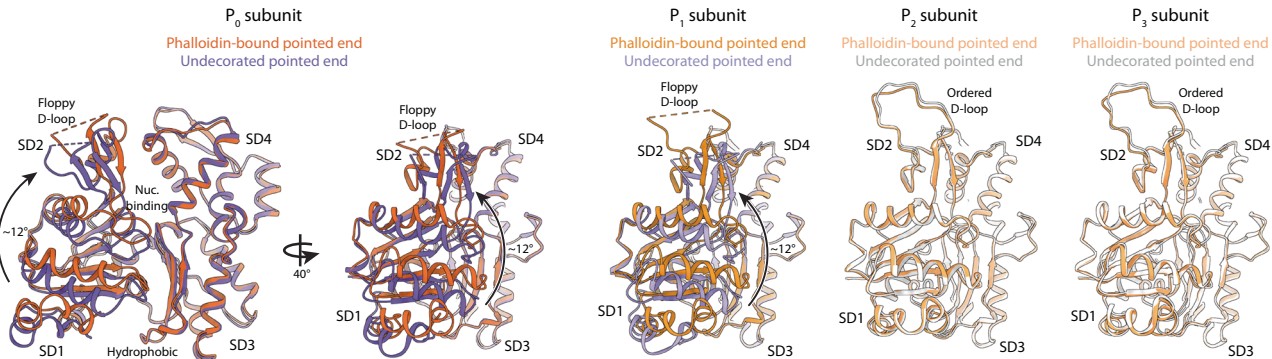

**Fig. 1 | Phalloidin bridges and flattens both terminal actin subunits at the pointed end. a** Exemplary 2D classes of the undecorated pointed end of actin filaments. Box size: 324 × 324 Å. **b** Cryo-EM density map of the undecorated pointed end of F-actin at a resolution of 3.1 Å. Actin subunits are labeled depending on their location on the filament, so that the ultimate subunit is $P_0$ and subunits towards the filament center have an increasing number ($P_1$, $P_2$, $P_3$). The ultimate ($P_0$) and penultimate ($P_1$) actin subunits are colored in purple and lilac, respectively, while the subunits towards the filament center ($P_2$ and beyond) are colored in light grey. The bound nucleotide is highlighted in yellow. **c** Cryo-EM density map of the phalloidin-bound pointed end of F-actin at a resolution of 3.7 Å. Phalloidin-bound actin subunits are colored from dark orange at the pointed end to lighter orange towards the filament center. The bound nucleotide and phalloidin are colored yellow and cyan, respectively. **d** Superimposition of the last four subunits ($P_0$-$P_3$) of the undecorated pointed end (purple and light grey) and the phalloidin-bound pointed end (shades of orange). The four actin subdomains (SD1-4), nucleotide binding site and hydrophobic cleft are annotated. The black arrow highlights the ~12° rotation of the outer subdomains (SD1, SD2) relative to the inner subdomains (SD3, SD4).

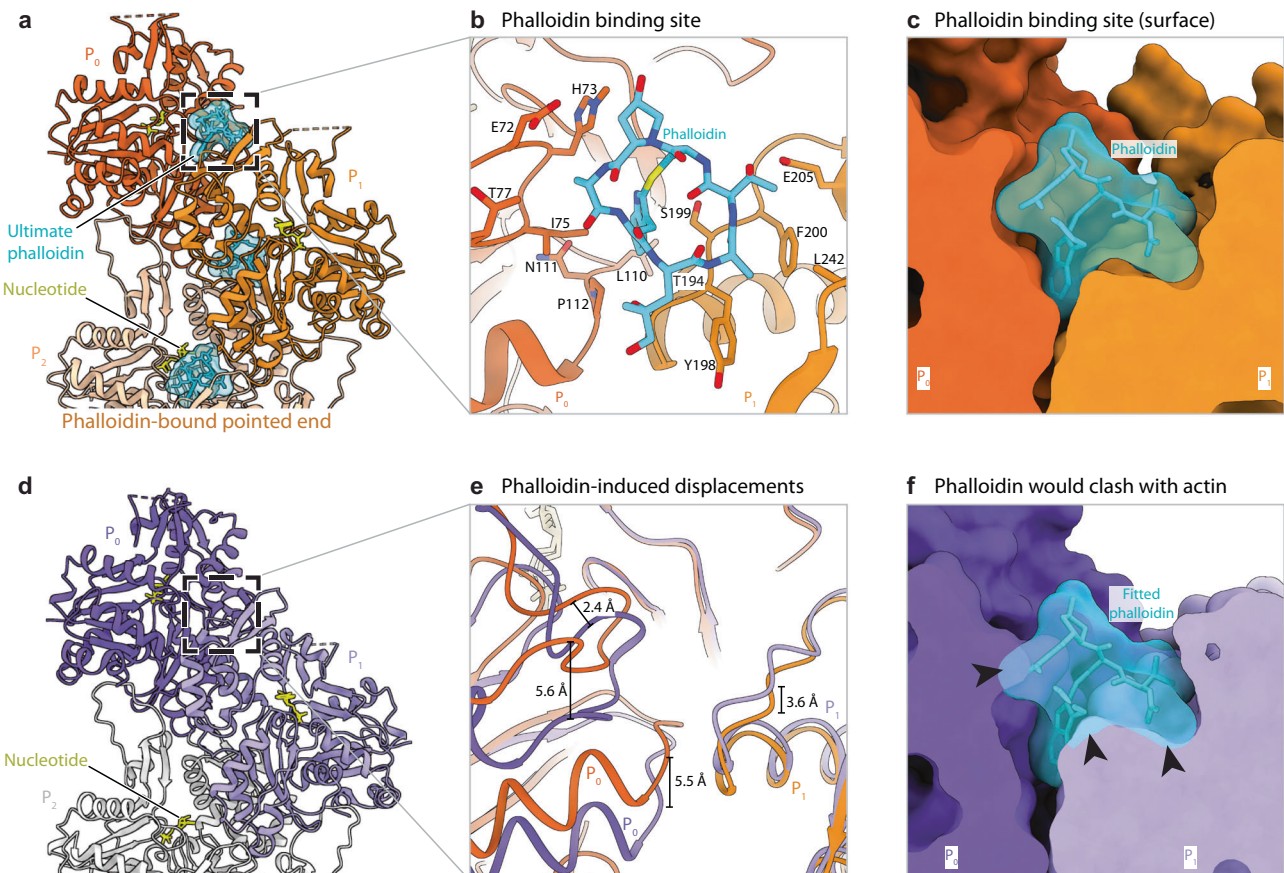

**Fig. 2 | Binding site of phalloidin to the ultimate and penultimate actin sub-units at the pointed end. a** Structure of the phalloidin-bound pointed end. Actin subunits are labelled from $P_0$ (ultimate actin subunit at the pointed end) to $P_2$ (actin subunit towards the filament center). Phalloidin-bound actin subunits are colored from dark orange at the pointed end to lighter orange towards the filament center. The bound nucleotide and phalloidin are colored yellow and cyan, respectively. **b** Zoom-in on the phalloidin binding site at the pointed end. The amino acids from subunits $P_0$ and $P_1$ that interact with phalloidin are annotated and shown with stick representation. **c** Clipped surface representation of the ultimate phalloidin binding site. **d** Structure of the undecorated pointed end. The ultimate ($P_0$) and penultimate ($P_1$) actin subunits are colored in purple and lilac, respectively, while the subunits towards the filament center ($P_2$ and beyond) are colored light grey. The bound nucleotide is highlighted in yellow. **e** Zoom-in on the interface between $P_0$ and $P_1$, superimposing the phalloidin-bound (orange) and undecorated (purple) pointed end. The phalloidin-induced displacement of various actin loops is indicated. **f** Surface representation of the undecorated pointed end with a fitted phalloidin in the ultimate phalloidin binding site. Phalloidin at this position would clash with actin.

internal subunits[27–29] (Figs. 1c, 2a–c). As described previously[28,29], phalloidin interacts with the proline rich loop (aa 108 – 112) and the sensor loop (aa 70 – 77) of SD1 of one subunit, the SD4-loop (aa 194–205) in the second subunit, and a small hydrophobic stretch involving I287 in SD3 of the third subunit (Fig. 2a, b, Supplementary Fig. 5, Supplementary Movie 1). We indeed observed phalloidin bound in such an arrangement between subunits $P_2$, $P_1$ and $P_0$ at the pointed end (penultimate phalloidin, Fig. 1c). Unexpectedly, we identified an additional phalloidin molecule between just the two last actin subunits ($P_1$ and $P_0$), directly demonstrating that the toxin can also stably occupy its ultimate binding site at the pointed end (Fig. 2a–c). While this binding site is not complete since the SD3 of a third actin subunit is missing, it enables phalloidin to act as a bridge that strengthens inter-subunit contacts at the pointed end of the filament. Strikingly, the two terminal actin subunits in the phalloidin-bound structure adopt a flattened conformation that resembles the general organization of subunits in the core of the filament, with the only exception that they display a disordered D-loop (Fig. 2a, Supplementary Fig. 4, Supplementary Movie 2). Thus, while phalloidin does not majorly affect the arrangement of the F-actin barbed end[31,32], it induces pivotal rearrangements at the pointed end by bridging and flattening the two terminal subunits, which rigidifies their arrangement. These conformational changes are required since phalloidin would otherwise sterically clash

with the twisted, G-actin-like terminal subunits (Fig. 2d–f). In conclusion, our findings explain in molecular detail how phalloidin prevents the pointed-end depolymerization of filaments by directly bridging the two terminal actin subunits, thereby stabilizing their arrangement at the filament end.

## Two DNase I molecules flexibly bind to the pointed end
Next, we aimed to elucidate how DNase I binds to the F-actin pointed end. In our previous work[31], we confirmed that DNase I remains stably associated with the pointed end of phalloidin-stabilized actin filaments[17,18]. We therefore developed a protocol in which we formed short β,γ-actin filaments in the presence of DNase I and phalloidin (see Methods). Image processing of all pointed-end particles revealed two additional densities compared to the undecorated pointed-end structures that were consistent with the size and shape of DNase I (Fig. 3a). We obtained a reconstruction of the F-actin–DNase I complex at a resolution of 3.6 Å (Fig. 3b, Supplementary Figs. 3, 6, 7a, Supplementary Table 2, Supplementary Movie 3).

The structure revealed that two DNase I molecules can simultaneously bind to the F-actin pointed end, with one DNase I predominantly interacting with the $P_1$ and the other with the $P_0$ subunit. However, the local resolution of both DNase I molecules was lower than that of F-actin in our reconstructions (Supplementary Figs. 6, 7a),

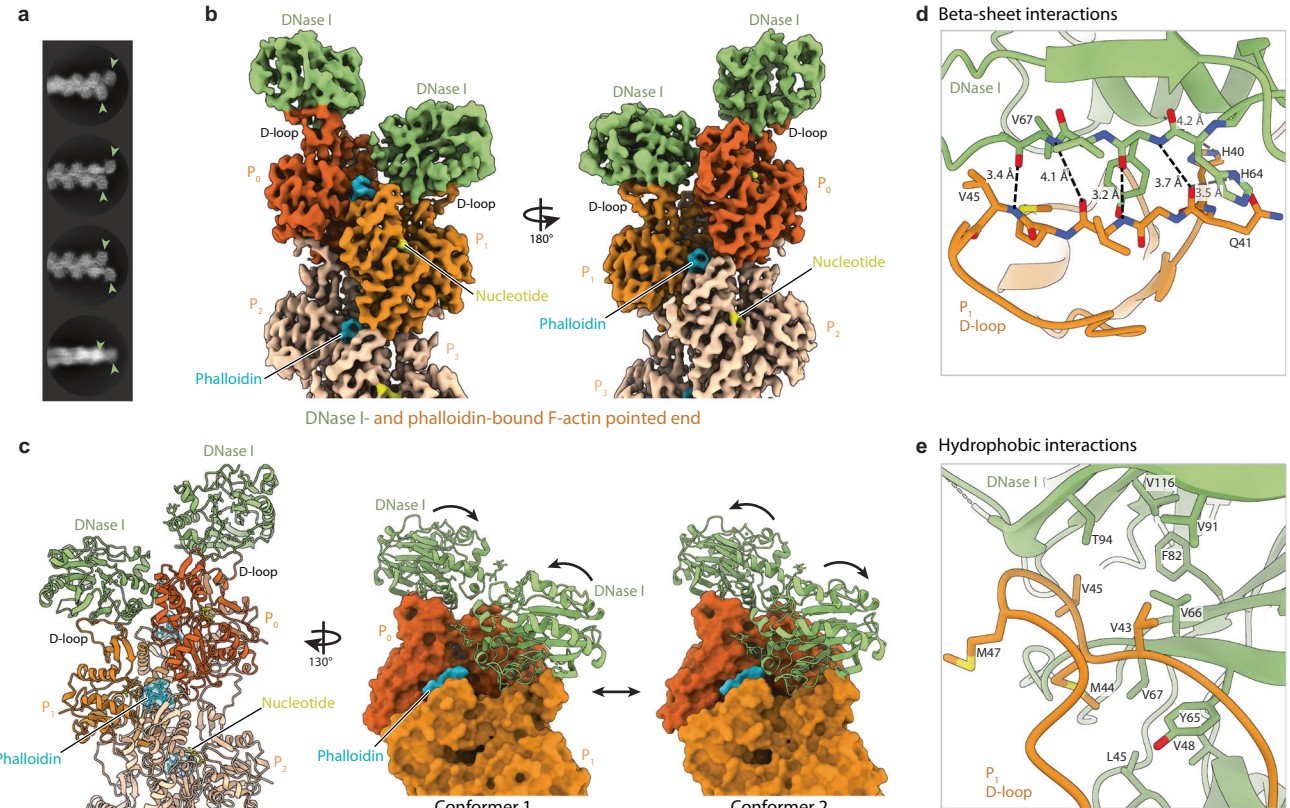

**Fig. 3 | Two DNase I molecules bind to the pointed end of actin filaments.**
**a** Exemplary 2D classes of the DNase I-bound pointed-end of actin filaments (conformer 1), stabilized with phalloidin. Light green arrowheads mark the position of the two DNase I molecules. Box size: 282 × 282 Å. **b** Cryo-EM density map of the DNase I-bound pointed end of F-actin at a resolution of 3.8 Å (conformer 1). Actin subunits are labelled from $P_0$ (ultimate actin subunit at the pointed end) to $P_2$ (actin subunit towards the filament center). Phalloidin-bound actin subunits are colored from dark orange at the pointed end to lighter orange towards the filament center. The two bound DNase I molecules, the bound nucleotide and phalloidin are colored light green, yellow and cyan, respectively. **c** Structure of the DNase I-bound pointed end of F-actin corresponding to conformer 1 (left). Structure of conformers 1 and 2 (modeled after maps obtained by 3D classification), see also the 3D variability results in Supplementary Movie 4 (middle and right). **d**, **e** Zoom-in on the two major interfaces between the D-loop of actin and DNase I (conformer 1).

suggesting that DNase I might display conformational flexibility. To further investigate this, we performed extensive 3D classifications and variability analyses (see Methods). This revealed that the D-loops of actin $P_1$ and $P_0$ can positionally fluctuate, which results in a 'wobbling' of the two DNase I molecules with respect to each other (Supplementary Movie 4). After isolating the particles that gave rise to the most different arrangements, we reconstructed two structures of the F-actin–DNase I complex at a resolution of ~3.8 Å, which we define as conformers 1 and 2 (Fig. 3a–c, Supplementary Fig. 7a, b). A comparison between the two conformers reveals that the position of the DNase I molecules on the pointed end can vary up to ~10 Å, with changes in binding angle up to ~10° (Fig. 3c, Supplementary Fig. 7b).

In both reconstructions, the conformation of actin subunits at the pointed end is highly similar to that in the phalloidin-stabilized pointed end structure without DNase I (Fig. 3c, Supplementary Fig. 4) with both terminal subunits remaining in a phalloidin-induced flattened conformation. Both DNase I molecules primarily engage with the D-loop in SD2 of the actin subunits, reminiscent of the interaction observed in G-actin–DNase I crystal structures[3,24] (Supplementary Fig. 7c–e). In addition, the interactions of the two DNase I molecules with the D-loops of actin $P_1$ and $P_0$ are comparable (Supplementary Fig. 7c, e). The main binding interface consists of a beta sheet-like arrangement with several hydrogen bond interactions between the backbones of the F-actin D-loop (aa 41 to 45) and residues 65 to 67 of DNase I (Fig. 3d). This interface is further strengthened by hydrophobic side chain interactions of D-loop residues V43, V45, M44 and M47 with surrounding residues from DNase I, which include L45, V48, Y65, V66,

V67, F82, V91, V94 and V116 (Fig. 3e). These interactions are comparable to those in G-actin–DNase I crystal structures (Supplementary Fig. 7c, d). We only noticed the loss of a minor DNase I-actin interaction, which is present in G-actin bound structures but not at the pointed end: Due to rearrangements in actin upon flattening, there are no interactions between actin SD4-residues T203 and E207 with the DNase I residues E13 and H44 (Supplementary Fig. 7c, d). However, the main interaction interfaces are retained, which supports the previous finding that DNase I displays similar affinities towards both G-actin and the F-actin pointed end[18]. Interestingly, the interface between DNase I and F-actin is comparable in both the conformers that we observe in our cryo-EM data. Therefore, we attribute the observed positional heterogeneity of the DNase I molecules to the flexibility of the D-loop of actin subunits $P_1$ and $P_0$. In conclusion, our structures reveal that two DNase I molecules bind to the F-actin pointed end in a flexible arrangement, which may be important to understand how DNase I depolymerizes filaments in the absence of phalloidin.

**Mechanism of DNase I-mediated disassembly of actin filaments**
Next, we asked how DNase I disassembles actin filaments under native conditions, i.e. without inter-subunit stabilization by phalloidin. To this end, we obtained a model of the native pointed end decorated by DNase I by aligning the pointed end structure with that of the F-actin–DNase I complex such that the SD2s of actin $P_1$ and $P_0$ from both structures superimposed (Fig. 4a). The alignment was performed by superimposing the resolved region of SD2 for both structures (aa 34-39, 50-69). The model shows that the outermost DNase I molecule can

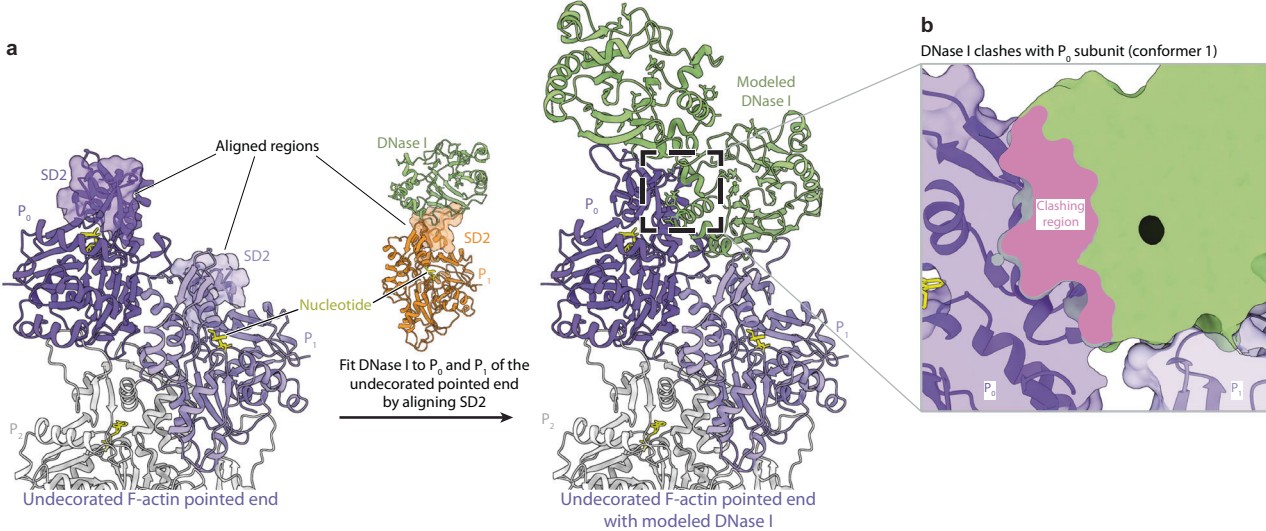

**Fig. 4 | DNase I bound to the penultimate actin subunit ($P_1$) displaces the ultimate actin subunit ($P_0$). a** Structure of the undecorated pointed end (left, purple and grey) and the penultimate subunit $P_1$ of the phalloidin-bound pointed end bound by DNase I (middle, orange and light green) highlighting subdomain 2 (SD2). The SD2s were used for the alignment of the two structures, yielding a model of the native pointed end decorated by DNase I (right). Actin subunits are labeled depending on their location on the filament, so that the ultimate subunit is $P_0$ and subunits towards the filament center have an increasing number ($P_1$, $P_2$, $P_3$). The bound nucleotide is highlighted in yellow. **b** Zoom-in image of the clashes between the penultimate DNase I and the actin subunit $P_0$. The clashing region is shown in pink.

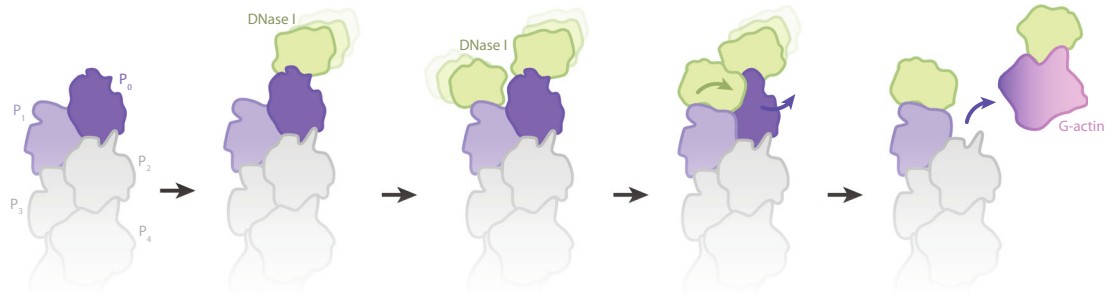

**Fig. 5 | Model of DNase I-mediated disassembly of actin filaments.** The model depicts the pointed end of an actin filament, with the flattened subunits in grey and the twisted subunits in shades of purple, the ultimate subunit is labeled $P_0$ and the subsequential actin subunits towards the filament center are labeled with increasing numbers ($P_1$-$P_4$). A DNase I molecule (depicted in light green) can bind to each of the last two actin subunits of the pointed end. DNase I binding to the penultimate subunit displaces the ultimate actin subunit from the pointed end, releasing a G-actin molecule (depicted in pink) bound to DNase I. For more a detail explanation we refer to the Results section.

readily bind to the D-loop of the ultimate subunit $P_0$ without introducing clashes or requiring conformational changes. Strikingly, however, the second DNase I, which interacts with the D-loop of the penultimate subunit $P_1$ would severely clash with the neighboring ultimate subunit $P_0$ (Fig. 4b, Supplementary Fig. 8). The model of the native pointed end decorated by DNase I therefore indicates that the binding of a DNase I molecule to the penultimate subunit could displace the ultimate one, leading to the destabilization of the inter-subunit actin packing and, ultimately, the loss of the ultimate subunit from the F-actin pointed end (Fig. 5). Based on this analysis, we propose the following mechanism for the filament depolymerization by DNase I (Fig. 5, Supplementary Movie 5): 1) At the F-actin pointed end, the D-loops of the two terminal actin subunits are flexible and available to form interactions; 2) The binding of a DNase I protein to actin $P_0$ results in an ordered D-loop, but does not induce changes to the pointed end arrangement; 3) A second DNase I molecule contacts the flexible D-loop of the penultimate $P_1$ subunit; 4) As this DNase I molecule forms a stable interaction with the $P_1$ D-loop, it leads to the dislocation of the $P_0$ subunit together with its bound DNase I from the pointed end; 5) The shorter pointed end now exposes actin $P_2$ and its D-loop, which can now interact with another DNase I molecule. These results highlight the importance of the sequential binding of two DNase I molecules to the F-actin pointed end in achieving both DNase I-driven depolymerization and the effective sequestration of depolymerized actin.

## Discussion

In this study, we examined the effects of phalloidin and DNase I on the conformation and stability of the pointed end of actin filaments. We show that phalloidin binds to the ultimate, incomplete binding site at the pointed end between the two terminal actin subunits, stabilizing the actin-subunit packing and inhibiting filament depolymerization. This is in contrast to the barbed end, where phalloidin interacts only with the penultimate and not with the ultimate, incomplete binding site, which is sufficient for stabilization[31]. This has also structural consequences. While phalloidin does not change the conformation of the barbed end, it alters the conformation and global arrangement of both terminal subunits at the pointed end. Hence, the binding site of phalloidin at the pointed end

and the barbed end is not identical, suggesting that phalloidin may have different affinities for the two ends.

Due to the conformational changes at the pointed end, phalloidin should be used with caution in future structural and biochemical studies on actin pointed ends, especially in combination with other pointed end-binding factors. The penultimate phalloidin binding site that lies between the three terminal actin subunits $P_2$, $P_1$, and $P_0$ overlaps with the binding site of the pointed-end capping protein tropomodulin[32,36] (Supplementary Fig. 9). Tropomodulin is the main ABP that binds the actin (thin) filament pointed end in muscle sarcomeres, where it controls thin filament lengths by preventing actin polymerization and disassembly at the pointed end[37,38]. Notably, both phalloidin and tropomodulin not only stabilize the pointed end arrangement by bridging the terminal subunits, but they also flatten the actin subunit $P_1$. Thus, these molecules appear to act by similar structural principles, despite their vast difference in size and nature. This suggests that also other proteins and small molecules that stabilize the pointed end of actin filaments might use a similar mechanism.

In contrast to phalloidin and tropomodulin, DNase I binding destabilizes the actin-subunit arrangement at the pointed end. Our results suggest that filament disassembly is not the result of direct repulsion between two DNase I molecules, as previously suggested[22]. Instead, the DNase I molecule bound to actin $P_1$ would severely clash with the ultimate actin $P_0$, indicating that filament depolymerization is the result of steric displacement. How does this DNase I-mediated actin filament disassembly mechanism compare to the mode-of-action of other F-actin depolymerases? The well-known actin-disassembly proteins of the ADF/cofilin family bind to the sides of F-actin to induce filament severing, suggesting a different mechanism. However, cofilin proteins synergize with cyclase-associated protein (CAP)[39-42], which forms oligomers and binds to the pointed end of cofilin-decorated F-actin[41,42]. A recent study suggested that two CAP subunits can simultaneously bind the $P_1$ and $P_0$ subunits at the pointed end, with accompanying molecular dynamics (MD) simulations predicting that CAP-binding destabilizes the packing between $P_1$ and $P_0$, leading to pointed-end depolymerization[41]. Although these findings have not yet been confirmed by structural data, they indicate that, similar to DNase I, CAP disassembles actin filaments by destabilizing subunit packing at the pointed end, with only one key difference. Whereas DNase I sequesters depolymerized actin subunits, CAP ensures efficient actin monomer recycling by nucleotide exchange for the polymerization of new filaments[41,43,44]. The similarities in depolymerization mechanism between DNase I and CAP are remarkable given their strong divergence in structure.

Finally, can we rationalize why both the association and dissociation rate constants of actin subunits at the pointed end are lower than those at the barbed end? Earlier low-resolution cryo-EM data and MD simulations predicted that unique contacts between the terminal subunits $P_1$ and $P_0$, mediated by the D-loop of $P_1$, impede subunit addition and loss at the pointed end[45,46]. However, recent high-resolution cryo-EM structures show that the D-loop of $P_1$ is disordered and not involved in such interactions[32,33], a finding we confirm (Fig. 1b, c), making these predictions inconsistent with experimental data.

A comparison between structures of undecorated F-actin ends reveals that all actin subunits at the barbed end display a flattened conformation[31,32,35], while the two terminal subunits at the pointed end adopt a twisted, G-actin like arrangement[32] (Fig. 1b, d). Based on this observation, it was recently proposed that actin subunits at the pointed end are conformationally 'primed for dissociation'[32,47]. However, actin dissociation rates at the pointed end have long been known to be lower compared to those at the barbed end[8-11], indicating that an alternative explanation is required. We thus carefully examined the conformational changes required for actin subunit addition or loss at either filament end (Fig. 6). Incorporation of a new monomer ($B_{-1}$) at the barbed end

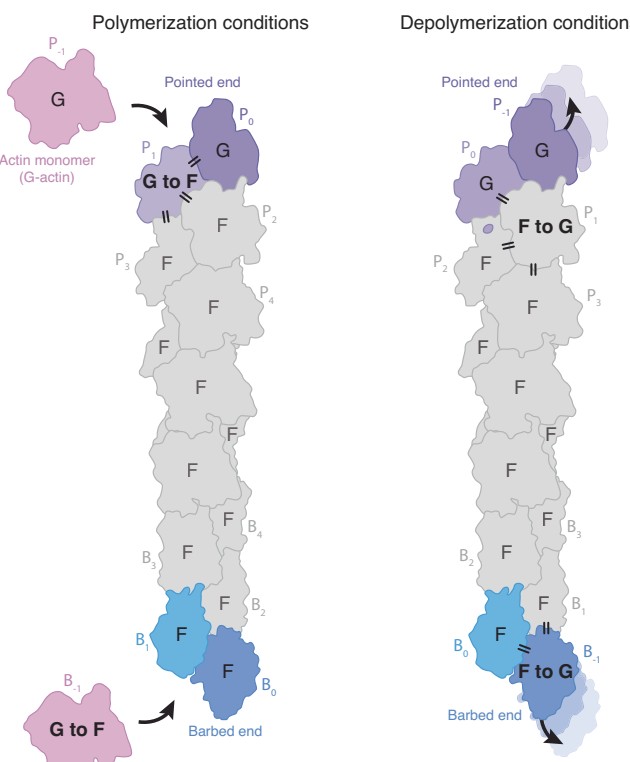

**Fig. 6 | Model explaining asymmetry in the incorporation and loss of subunits at opposite filament ends.** Actin monomers are colored in pink. The internal subunits of actin filaments are colored in grey, whereas the last two subunits of the barbed end and the pointed end are colored in shades of blue and purple, respectively. Subunits of the pointed end are labeled $P_x$, with x starting from 0 with increasing numbers towards the filament center, and subunits at the barbed end are labeled in a similar manner with $B_x$. Subunits labeled G are in a twisted, G-actin like conformation, and subunits labeled F are in a flattened, filament-like conformation. Subunits labeled G to F and F to G are those that are expected to undergo a conformational change during polymerization and depolymerization, respectively. For a detailed discussion of this model we refer to the Discussion section.

requires it to form contacts with the two terminal subunits ($B_1$ and $B_0$) and flatten in the process. Similarly, at the pointed end, the incoming monomer ($P_{-1}$) needs to interact with the two terminal subunits ($P_1$ and $P_0$), but can itself remain in a twisted, G-actin-like state. It is the penultimate subunit $P_1$, which has to flatten to accommodate the incoming monomer (Fig. 6). Hence, addition of a new monomer to the filament invariably requires the flattening of one actin subunit regardless of where it is added. Importantly, however, the subunits that must flatten at the barbed or pointed end are non-equivalent. The subunit undergoing this conformational change at the barbed end – the incoming subunit $B_{-1}$ – can do so without restraints by interactions with other subunits. At the pointed end on the other hand, flattening of the $P_1$ subunit is restrained by interactions with four other actin subunits in the filament. Similar principles apply to the process of subunit loss, in which the subunit dissociating from the barbed end ($B_{-1}$) can freely twist, whereas the internal subunit undergoing the same conformational change at the pointed end ($P_1$) is conformationally more restricted (Fig. 6). Indeed, the interaction interfaces between SD1 of $P_1$ and the D-loop of $P_3$, those between the proline rich loop of $P_1$ and SD4 of $P_2$, and those between SD2 of $P_1$ and the H-plug (SD4) of $P_0$ (Supplementary Fig. 10) all need to rearrange to enable actin subunit flattening or twisting. Hence, we propose that the conformational G-to-F transition impedes monomer addition and loss at the pointed, but not at the barbed end, explaining their kinetic asymmetry.

## Methods

### Purification of rabbit skeletal α-actin

Native skeletal α-actin was purified from rabbit-muscle acetone powder as described previously[7]. We thawed 0.5 g of acetone powder and dissolved it in 10 ml G-buffer (5 mM Tris pH 7.5, 0.2 mM CaCl$_2$, 0.2 mM ATP, 0.5 mM TCEP, 0.1 mM NaN$_3$) by stirring at 4 °C for 25 min. After filtering the suspension and collecting the supernatant, the insoluble material was recovered from the filter paper and dissolved in 10 ml G-buffer through the same procedure. The mixture was filtered again and the supernatant was combined with the supernatant collected earlier to a total volume of 20 ml. It was then subjected to ultracentrifugation at 100,000 g for thirty minutes in a Ti70 rotor to remove any remaining insoluble material. The supernatant was collected and actin was polymerized at room temperature for 1 hour by the addition of 2 mM MgCl$_2$ and 100 mM KCl (final concentrations). To further remove proteins bound to F-actin, we added solid KCl to the solution (final concentration 800 mM). Following another 1-hour incubation at room temperature, the sample was ultracentrifuged at 100,000 g for 2 h to pellet all actin filaments. The filaments were resuspended in 5 ml G-buffer and actin was depolymerized by dialysis in 1 L G-buffer for two days. Then, the dialyzed sample was ultracentrifuged at 100,000 g for 30 min to remove any remaining filaments and the supernatant containing G-actin was recovered. To ensure the removal of all impurities, the protein was subjected to another 2-day procedure of actin polymerization, 800 mM KCl-wash and depolymerization in G-buffer by dialysis. Finally, purified G-actin was flash frozen in liquid nitrogen and stored at −80 °C.

### Purification of human cytoplasmic β-actin

Human cytoplasmic β-actin was expressed recombinantly in BTI-Tnao38 insect cells (Provider: A. Musacchio, MPI Dortmund, CVCL_Z252) as a fusion protein[7,31,48], with the ABP thymosin β4 and a His10-tag fused to the C-terminus. The fused thymosin β4 prevents actin from polymerizing. The protein additionally contained the C272A substitution, because residue C272 is prone to oxidation in aqueous solutions[49]. In human and rabbit skeletal α-actin, the equivalent residue is also alanine.

The fusion protein was expressed by infecting the BTI-Tnao38 cells with baculovirus. Forty-eight hours later, the cells were harvested by centrifugation (5000 g, 10 min) and stored at −80 °C until further use. To purify the protein, the cells were thawed, resuspended in Lysis buffer (10 mM Tris pH 8, 50 mM KCl, 5 mM CaCl$_2$, 1 mM ATP, 0.5 mM TCEP and cOmplete protease inhibitor (Roche)) and lysed using a fluidizer. All insoluble materials were then removed by ultracentrifugation (100,000 g, 25 min) and the supernatant was filtered and loaded onto a 5 ml HisTrap FF crude column using a peristaltic pump. After washing with Lysis buffer supplemented with 20 mM imidazole for ~50 ml, actin-thymosin-β4-His10 was eluted from the column using an imidazole gradient (20 mM to 160 mM in 10 min). The eluted fusion-protein sample was dialyzed overnight in G-buffer to decrease the salt concentration. The next day, it was incubated with chymotrypsin (weight ratio of 1/250 chymotrypsin/actin) at 25 °C for 20 minutes to separate β-actin from the thymosin-β4-His10 tag. Afterwards, 0.2 mM PMSF (final concentration) was added to terminate the proteolysis reaction, and the sample was applied to a clean 5 ml HisTrap FF crude column. The flowthrough, which contained β-actin, was collected and β-actin was polymerized by the addition of 2 mM MgCl$_2$ and 100 mM KCl (final concentrations) for 2 hours at room temperature, followed by incubation at 4 °C overnight. The next morning, the formed actin filaments were pelleted by ultracentrifugation (210,000 g, 2 hours). The filaments were resuspended and depolymerized by dialysis in G-buffer for 2 days. Following a final round of ultracentrifugation to pellet any remaining filaments or aggregates, the supernatant containing monomeric β-actin was collected was flash frozen in liquid nitrogen and stored at −80 °C.

### Purification of bovine cytoplasmic β-actin and γ-actin

A mixture of native cytoplasmic β-actin and γ-actin, from now on referred to as β,γ-actin, was purified from bovine thymus as described in refs. [34,35,50]. Bovine thymus was severed into small fragments and was homogenized in a precooled blender together with ice cold Holo-Extraction buffer (10 mM Tris-Cl pH 8.0, 7.5 mM CaCl$_2$, 1 mM ATP, 5 mM β-mercaptoethanol, 0.03 mg/ml benzamidine, 1 mM PMSF, 0.04 mg/ml trypsin inhibitor, 0.02 mg/ml leupeptin, 0.01 mg/ml pepstatin, 0.01 mg/ml apoprotein). Afterwards, additional 2.5 mM β-mercaptoethanol was added to the lysate and the pH was checked and readjusted to pH 8.0 if necessary. After initial centrifugation the lysate was filtered through a nylon membrane [100 μm] and hard spun in an ultracentrifuge. The volume of the cleared supernatant was measured out and the salt and the imidazole concentrations were adjusted (KCl to 50 mM, imidazole to 20 mM). The supernatant was incubated with the gelsolin G4-6 fragment to promote the formation of actin:gelsolin G4-6 complexes. To this end, 4 mg of 10xhis-gelsolinG4-6 were added for each g of thymus to the lysate and dialyzed into IMAC wash buffer overnight (10 mM Tris-Cl pH 8.0, 50 mM KCl, 20 mM imidazole, 5 mM CaCl$_2$, 0.15 mM ATP, 5 mM β-mercaptoethanol). The lysate containing the actin:gelsolin G4-6 complex was then circulated over a Ni$^{2+}$ superflow column. Actin monomers were eluted with Elution Buffer (10 mM Tris-Cl pH 8.0, 50 mM KCl, 20 mM imidazole, 5 mM EGTA, 0.15 mM ATP, 5 mM β-mercaptoethanol) into a collection tray containing MgCl$_2$ (2 mM final concentration). Actin containing fractions were identified by gelation, pooled and further polymerized for 4 h at RT after adjusting to 1xKMEI and 0.5 mM ATP. After ultracentrifugation, the actin filament pellet was resuspended in F buffer (1xKMEI, 1xG-Buffer) and stored in continuous dialysis at 4 °C. F buffer containing fresh ATP and TCEP was continuously exchanged every 4 weeks. The purified β,γ-actin was kept in its filamentous form by continuous dialysis in 1xKMEI buffer (0.5 mM ATP, 1 mM TCEP, 50 mM KCl, 1.5 mM MgCl$_2$, 1 mM EGTA, 10 mM imidazole pH 7) at 4 °C, with exchanges to fresh dialysis buffer every four weeks.

### Purification of the G-actin – DNase I complex

The purification of the G-actin – DNase I complex was described previously in ref. [31]. DNase I from bovine pancreas (DNase I, Serva, cat. no. 18535.02) was dissolved at 20 mg ml$^{-1}$ (~ 666 μM) in G-buffer containing 15 μg/ml benzamidine and 1 mM PMSF. To form a complex with actin, we mixed 800 μl of DNase I (666 μM) with 3 ml of filamentous native bovine β,γ-actin (89 μM) and dialyzed the mixture in G-buffer containing 15 μM benzamidine and 1 mM PMSF for four days to ensure complete actin depolymerization. Then, the mixture was subjected to ultracentrifugation in a TLA110.1 rotor (Beckman) for thirty minutes to remove any remaining filaments or aggregates. The supernatant was collected and the G-actin – DNase I complex was purified using size-exclusion chromatography over a Superdex 200 16/600 column in G-buffer. Fractions containing the DNase I: β,γ-actin complex were collected and further concentrated to a final concentration of approximately 260 μM and stored at 4 °C for up to three months.

### Purification of formins

The FH2 domain (residues 972 – 1390) of *S. pombe* Cdc12 (termed Cdc12(FH2)) was purified as described in ref. [35]. The protein was expressed with an N-terminal His10-SUMO3 tag in *E. coli* BL21 Star pRARE cells at 18 °C for 16 h. The *E. coli* cells were pelleted, resuspended in Lysis buffer (50 mM NaPO$_4$ pH 8.0, 400 mM NaCl, 0.75 mM β-mercaptoethanol, 15 μg/ml benzamidine, 1x complete protease inhibitors, 1 mM PMSF, DNase I) and lysed using a high-pressure homogenizer (Emulsiflex). His10-SUMO3-Cdc12(FH2) was then purified by IMAC using a 5 ml HiTrap Chelating column and eluted in a gradient with imidazole-containing Elution buffer (50 mM NaPO$_4$ pH 7.5, 400 mM NaCl, 400 mM imidazole, 0.5 mM β-mercaptoethanol). The His10-Sumo tag was cleaved using SenP2 protease overnight and the sample was desalted

into Lysis buffer over a HiPrep 26/10 column. After re-circulating over a 5 ml HiTrap Chelating column, the flowthrough containing Cdc12(FH2) was concentrated and subjected to size-exclusion chromatography over a Superdex 200 16/60 column into Storage buffer-1 (20 mM HEPES pH 7.5, 200 mM NaCl, 0.5 mM TCEP). Fractions containing pure Cdc12(FH2) were pulled, concentrated, supplemented with 20% (v/v) glycerol and flash frozen in liquid nitrogen.

The tandem FH1-FH2 domain and complete C-terminus (residues 469 – 1240) of human inverted formin-2 isoform 2 (termed INF2(FH1FH2C)) was purified as described[35]. The protein was expressed with an N-terminal His6-Z-TEV-Gly5 tag in *E. coli* BL21 Star pRARE cells at 18 °C for 16 h. The Z-tag is a solubility tag based on Protein A from *S. aureus*, while Gly5 stands for penta-glycine. The *E. coli* cells were pelleted, resuspended in Lysis buffer-2 (50 mM NaPO$_4$ pH 8.0, 400 mM NaCl, 10 mM arginine, 10 mM glutamic acid, 0.75 mM β-mercaptoethanol, 15 μg/ml benzamidine, 1x complete protease inhibitors, 1 mM PMSF, DNase I) and lysed using a high-pressure homogenizer. His6-Z-TEV-Gly5-INF2(FH1FH2C) was then purified by IMAC using a 5 ml HiTrap Chelating column and eluted in a gradient with imidazole-containing Elution buffer-2 (50 mM NaPO$_4$ pH 7.5, 400 mM NaCl, 400 mM imidazole, 10 mM arginine, 10 mM glutamic acid, 0.5 mM β-mercaptoethanol). The His6-Z-tag was cleaved using TEV protease overnight and the sample was desalted into Lysis buffer over a HiPrep 26/10 column. After re-circulating over a 5 ml HiTrap Chelating column, the flowthrough containing Gly5-INF2(FH1FH2C) was concentrated and subjected to size-exclusion chromatography over a Superdex 200 16/60 column into storage buffer-2 (25 mM HEPES pH 7.5, 200 mM KCl, 5 mM arginine, 5 mM glutamic acid, 0.5 mM TCEP). Fractions containing pure Gly5-INF2(FH1FH2C) were pulled, concentrated, supplemented with 20% (v/v) glycerol and flash frozen in liquid nitrogen.

### Purifcation of tropomodulin-3

The purification of full-length tropomodulin-3 (Tmod3) was described previously[35]. The protein was expressed with an N-terminal His6-Z-TEV-Gly5 tag in *E. coli* BL21 Star pRARE cells at 18 °C for 16 h. The *E. coli* cells were pelleted, resuspended in Lysis buffer-3 (20 mM Tris-Cl pH 7.0, 300 mM NaCl, 0.75 mM β-mercaptoethanol, 15 μg/ml benzamidine, 1xcomplete protease inhibitors, 1 mM PMSF, DNase I) and lysed using a high-pressure homogenizer. His6-Z-TEV-Gly5-Tmod3 was then purified by IMAC using a 5 ml HiTrap Chelating column and eluted in a gradient with imidazole-containing Elution buffer-3 (50 mM NaPO$_4$ pH 7.5, 300 mM NaCl, 400 mM imidazole, 0.75 mM β-mercaptoethanol). The His6-Z-tag was cleaved using TEV protease overnight and the sample was desalted into Lysis buffer over a HiPrep 26/10 column. After re-circulating over a 5 ml HiTrap Chelating column, the flowthrough containing Gly5-Tmod3 was concentrated and subjected to size-exclusion chromatography over a Superdex 200 16/60 column intoStorage buffer-3 (20 mM HEPES pH 7.0, 50 mM NaCl, 0.5 mM TCEP). Fractions containing pure Gly5-Tmod3 were pulled, concentrated, supplemented with 20% (v/v) glycerol and flash frozen in liquid nitrogen.

### Sample preparation for cryo-EM

The structural investigation of the ends of F-actin requires the generation of short filaments, so that enough ends can be imaged per micrograph to enable high-resolution structure determination. In order to maximize the structural insights from our cryo-EM data, we designed our experiments so that we could obtain relevant structural information on both the pointed and barbed end of F-actin. For the latter, we focused on how formins Cdc12 and INF2 bind the barbed end to regulate actin-subunit addition. The structural analyses of these data on the F-actin barbed end are described elsewhere[35]. Thus, the sample preparation protocols that enabled us to determine the undecorated and phalloidin-stabilized pointed end structures were also described previously in ref. [35].

### Undecorated pointed end cryo-EM sample preparation

As previously described in ref. [35], 43 μM of G-actin (native rabbit skeletal α-actin) was diluted to 8 μM in G-Mg buffer (2 mM HEPES pH 7.1, 0.5 mM TCEP, 0.2 mM ATP, 0.1 mM MgCl$_2$) on ice for five minutes. Then, actin was polymerized by the addition of 1/10$^{th}$ of volume of 10xKMEH (100 mM HEPES pH 7.1, 1 M KCl, 20 mM MgCl$_2$, 10 mM EGTA) and INF2(FH1FH2C), resulting in final concentrations of ~7 μM and ~4.5 μM for actin and INF2(FH1FH2C), respectively. The sample was polymerized at room temperature for fifteen minutes after which it was directly used for cryo-EM grid freezing.

### Phalloidin-bound pointed end cryo-EM sample preparation

As previously described[35], G-actin (recombinant human cytoplasmic β-actin) was mixed with Cdc12(FH2) on ice to final concentrations of 30 μM and 16 μM, respectively, in a volume of 50 μl G-buffer. The sample was then incubated with 5.6 μl of 10xME (500 μM MgCl$_2$, 2 mM EGTA) for five minutes. Afterwards, actin was polymerized by the addition of 6.2 μl polymerization buffer (64 mM HEPES pH 7.1, 640 mM KCl, 12.8 mM MgCl$_2$, 6.4 mM EGTA, 100 μM tropomodulin-3, 600 μM phalloidin, 12% DMSO) for two minutes at room temperature, followed by fifteen to twenty minutes on ice. The formed short actin filaments were pelleted by ultracentrifugation at 400,000 g for twenty minutes in a TLA120.1 rotor (Beckman). After removing the supernatant, the pellet was resuspended in 50 μl of Resuspension buffer (10 mM HEPES pH 7.1, 100 mM KCl, 2 mM MgCl$_2$, 1 mM EGTA, 0.5 mM TCEP, 150 nM Cdc12(FH2), 150 nM tropomodulin-3, 20 μM phalloidin) and used for cryo-EM grid freezing.

### DNase I-bound pointed end cryo-EM sample preparation

For the preparation of the Cryo-EM sample, the following components (final concentrations) were mixed in G-Buffer to a final volume of 50 μL: 50 μM DNase I: β,γ-actin; 10 μM β,γ-actin; 20 μM INF2(FH1FH2C). This mixture was supplemented with 5,5 μl 10xME (500 μM MgCl$_2$, 2 mM EGTA) and incubated for 2 minutes on ice. Actin polymerization was triggered by addition of 6.2 μl 10xKMEI +phalloidin (455 μM phalloidin, 5 mM ATP, 10 mM TCEP, 500 mM KCl, 15 mM MgCl$_2$, 10 mM EGTA, 100 mM imidazole pH 7). The sample was incubated for 5 minutes at room temperature and 30 minutes on ice, and then ultracentrifuged in a TLA100 rotor at 280,000 g for 20 min. The supernatant was discarded and the pelleted short filaments were resuspended in 50 μL of 1xGF buffer + 10 μM phalloidin and directly used for cryo-EM.

### Cryo-EM grid preparation and screening

Grids were prepared and frozen as described previously[35]. Briefly, 2.8 μl of each sample was applied to a R2/1 Au 200 mesh holey-carbon grid (Quantifoil) which had been previously glow-discharged. A Vitrobot Mark IV (Thermo Fisher Scientific) was used to blot excess solution and freeze the grids in liquid ethane or a liquid ethane/propane mixture. This process was done at 13 °C and 100% humidity inside the Vitrobot chamber. Blotting forces of 0 to 2, and a blotting time of 3 seconds were used.

Grids were screened on a 200 kV Talos Arctica cryo-microscope (Thermo Fisher Scientific) with a Falcon III direct electron detector (Thermo Fisher Scientific) in linear mode. Grids were screened in low and high magnification using EPU (Thermo Fisher Scientific). Filament length (ideally 30 −150 nm) was optimized during sample preparation so that as many ends as possible were visible per micrograph. Grids with short filaments and a good ice thickness distribution were stored in liquid nitrogen and later transferred to a Titan Krios microscope for data collection.

### Cryo-EM data collection and preprocessing

The dataset that yielded the undecorated pointed end structure without phalloidin was collected on a 300 kV Titan Krios G3

microscope (Thermo Fisher Scientific) with a K3-direct electron detector (Gatan) and a BioQuantum post-column energy filter (Gatan, slit-width of 15 eV). Movies were collected in super-resolution mode using EPU (Thermo Fisher Scientific), at a pixel size of 0.45 Å (0.9 Å after 2-fold binning during motion correction).

The datasets that yielded the phalloidin-stabilized pointed end structures with and without bound DNase I were collected on a 300 kV Titan Krios G2 microscope (Thermo Fisher Scientific) with an in-column Cs corrector, a K3-direct electron detector and a BioQuantum post-column energy filter (slit-width of 15 eV for both datasets). Movies were collected in super-resolution mode using EPU (Thermo Fisher Scientific), at a pixel size of 0.44 Å (0.88 Å after 2-fold binning during motion correction).

All datasets were monitored and preprocessed on the fly using TranSPHIRE[51]. Within TranSPHIRE, camera-gain and beam-induced motion were corrected using UCSF MotionCor2 v1.3.0[52]. MotionCor2 was also used for the two-fold binning of the super-resolution collected movies, while CTFFIND4.13[53] was used to estimate the contrast transfer function (CTF) of each micrograph. Filament ends were picked using SPHIRE-crYOLO[54] through an approach that was previously described[35].

### Image processing of the undecorated pointed end dataset
For the undecorated F-actin pointed end, we collected a large dataset comprising 20,305 micrographs. A total of 1,935,707 particles were picked with crYOLO, which were binned 3 times and extracted in RELION with a box size of 120 × 120 pixels and a pixel size of 2.7 Å. These particles were imported into CryoSPARC and subjected to an initial round of 2D classification to select all filament or filament end particles. The good particles were used to perform a 3D Hetero-geneous Refinement using four references: the F-actin pointed end, the F-actin barbed end, the complete filament core and a formin-bound barbed end. The 241,935 particles corresponding to the pointed end of F-actin were further 2D classified and selected, and the resulting particles were non-uniformly refined. These particles were unbinned in RELION (360 × 360 pixel box and 0.9 Å pixel size) and afterwards non-uniformly refined in CryoSPARC yielding a den-sity map of the F-actin pointed at a resolution of 3.5 Å. The resulting particles were re-imported into RELION for Bayesian Polishing and CTF refinement. This was performed to improve the estimations of beam-induced particle motions (Bayesian Polishing) and to estimate per particle defocus values and for microscope-aberration correction (CTF refinements). The polished particles were imported in CryoS-PARC, 2D classified and non-uniformly refined. We then created a soft mask (1 pixel extension, 12 pixels soft edge) around the four terminal subunits of the pointed end. To improve the density of the terminal subunits, we performed a final local refinement with this mask, which yielded a reconstruction of the undecorated pointed end of F-actin at a resolution of 3.1 Å.

### Image processing of the phalloidin-bound pointed end dataset
We collected 20,393 micrographs for the phalloidin-stabilized poin-ted end dataset. 5,200,600 particles were picked using crYOLO. These particles were extracted in a 120×120 pixels box in RELION (4x binned, resulting pixel size 3.52 Å). After importing them into CryoSPARC, we first performed a 3D Heterogenous Refinement with three references: 1) the Cdc12-bound F-actin barbed end, 2) a com-plete filament center and 3) the F-actin pointed end. Interestingly, tropomodulin-3 was present in this sample but no pointed-end par-ticles were observed to be bound by it. This indicates that tropomodulin-3 was displaced by phalloidin since they compete for an overlapping binding site (See Discussion, Supplementary Fig. 9). The 926,619 particles that refined into the pointed end class were selected and subjected to two rounds of 2D classification to remove junk particles. The remaining 626,524 particles were non-uniformly refined to a resolution of 7.1 Å (bin 4 Nyquist). After particle un-

binning (480 ×480 pixel box, resulting pixel size 0.88 Å), another non-uniform refinement yielded a density map at a resolution of 3.7 Å. Then, the particles were 3D classified without image alignment (removing 212,201 particles) and converted to RELION for Bayesian Polishing and CTF refinements. The particles were then 3D classified with image alignment in RELION into four classes. While two of the classes were junk and one was empty, ~83% of the particles classified into a single good class. Following particle-duplicate removal, the remaining 280,802 particles were re-imported into CryoSPARC and non-uniformly refined to a resolution of 3.64 Å. We then created a soft mask (1 pixel extension, 12 pixels soft edge) around the four terminal subunits of the pointed end. To improve the density of the terminal subunits, we performed a final local refinement with this mask, which yielded a reconstruction of the phalloidin-stabilized pointed end of F-actin at 3.7 Å.

### Image processing of the DNase I-bound pointed end dataset
We collected a dataset of 15,977 micrographs. Using crYOLO, we picked 4,418,492 particles, which were extracted in RELION with 4x binning at a pixel size of 3.52 Å and with a box of 120 ×120 pixels. These particles were imported into CryoSPARC and subjected to an initial round of 2D classification to select all filament or filament end particles. The good particles were used to perform a 3D Heterogeneous Refinement using four references: the F-actin DNase I-bound pointed end, the F-actin undecorated barbed end, the complete filament core and a formin-bound barbed end. The DNase I-bound pointed end particles were subjected to 2D classification to remove junk particles, and the resulting 714,830 particles were non-uniformly refined. The particles were con-verted back to RELION and 3D classified, which revealed 4 classes: a class with bad particles, and three classes with DNase I bound at the pointed end, one of which was shifted on the z axis by one actin subunit. All particles were unbinned and the shifted particles were re-centered in z by 27.5 Å (value corresponding to the helical rise of actin subunits). The resulting 618,802 particles were subjected to Bayesian polishing and then re-imported to CryoSPARC for non-uniform refinement, which yielded a density map at 3.55 Å. In this map, the actin is well resolved, but the DNase I density is weak and displays a lower local resolution. To attempt to get better DNase I densities, we created a mask for further 3D classifications. This mask encompasses the two bound DNase I mole-cules and was created with a 1 pixel extension and a 12 pixels soft edge. A 3D classification using this mask as a focus mask revealed four classes, where the two major classes presented better defined densities for both DNase I molecules. The particles corresponding to these two classes were processed separately: first non-uniformly refined and then locally refined using a mask encompassing the two DNase I molecules and last four actin subunits. The local refinements yielded two density maps at 3.8 Å resolution.

The 3D variability analysis was performed in cryoSPARC using classes 0, 1 and 2 of the last 3D classification (a total of 490,879 par-ticles). Class 3 was discarded because it has an extra actin subunit, meaning that it is shifted in the helical rise by 27.5 Å. This analysis was run using a mask that comprises the two DNase I molecules and the last 4 actin subunits of the pointed end. To display the results, we ran the CryoSPARC 3D Variability Display job. The resulting volume series of 20 frames for each of the three solved orthogonal modes were visua-lized using UCSF ChimeraX[55].

### Model building of F-actin pointed end dataset
Four alpha F-actin subunits from a previously published model of F-actin in the Mg$^{2+}$-ADP nucleotide state were used as an initial refer-ence for model building (PDB: 8A2T)[7]. Each subunit was individually rigid-body fitted in the cryo-EM density map using UCSF ChimeraX-v1.6. The only regions that did not fit in the density were the SD1 and SD2 of the ultimate and penultimate actin subunits. For this reason, four regions in the last two actin subunit were split and rigid-body

fitted into the cryo-EM density map independently: (1) D-loop, residues 34-69; (2) Top SD1 helix close to the D-loop, residues 77-96; (3) Central SD1 helix, residues 11-128; and (4) C-terminal region from SD1 (348-375). The model was manually adjusted in coot[56] first and then iteratively refined by manual model building in coot and phenix real-space refine with applied geometric restraints[57].

## Model building of the phalloidin-bound pointed end dataset

The initial model of beta F-actin bound to phalloidin was retrieved from a previously published structure of the phalloidin- and Cdc12-bound barbed end of F-actin (PDB: 8RTT)[35]. Four actin subunits from this model were individually rigid-body fitted in the cryo-EM density map using UCSF ChimeraX-v1.6. The model was manually adjusted in coot first and then iteratively refined by manual model building in coot and phenix real-space refine with applied geometric restraints.

## Model building of the DNase I-bound pointed end dataset

The initial model for the phalloidin-stabilized pointed end was retrieved from a previously published structure of the phalloidin-stabilized F-actin (PDB: 6T20) and the DNaseI structure was retrieved from a model derived from X-ray diffraction (PDB: 2A42). Four actin subunits and two DNase I molecules were rigid-body fitted in the cryo-EM density map using UCSF ChimeraX-v1.6. The model was manually adjusted in coot first and then iteratively refined by manual model building in coot and phenix real-space refine with applied geometric restraints.

### Reporting summary

Further information on research design is available in the Nature Portfolio Reporting Summary linked to this article.

## Data availability

The cryo-EM maps generated in this study have been deposited in the Electron Microscopy Data Bank (EMDB) under accession codes (dataset in brackets): EMD-50507 (F-actin pointed end), EMD-50506 (phalloidin-bound F-actin pointed end), EMD-50516 (phalloidin- and DNase I-bound F-actin pointed end, maps before 3D classification and conformer 1), EMD-50517 (phalloidin- and DNase I-bound F-actin pointed end, conformer 2). Sharpened and unsharpened maps, unfiltered half-maps and the masks used for refinements are included in each entry. Associated protein models have been deposited in the Protein Data Bank (PDB) with accession codes 9FJO (F-actin pointed end), 9FJM (phalloidin-bound F-actin pointed end), 9FJU (phalloidin- and DNase I-bound F-actin pointed end, conformer 1), 9FJY (phalloidin- and DNase I-bound F-actin pointed end, conformer 2). The previously published structures that were used to build the initial models for the structures presented in this article can be found in the PDB under the accession codes: 8A2T, 8RTT, 6T20, 2A42. Source Data is provided as a Source Data file.

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

## Acknowledgements
We thank S. Bergbrede and S. Koch for overall wet lab support and for assistance with protein purifications. This work was supported by funds from the Max Planck Society (to P.B., and S.R.), the German Research Foundation (DFG, grant no. BI 1998/2-1 to P.B.) and the European Research Council under the European Union's Horizon 2020 Programme (ERC-2019-SyG, grant no. 856118 to S.R) W.O. and M.B.S. are supported by postdoctoral fellowships from the Alexander von Humboldt foundation.

## Author contributions
S.R. and P.B. conceived and supervised the study. M.B.S., W.O., S.R. and P.B. designed the experiments. P.B., M.B.S. and W.O. purified proteins. M.B.S., W.O. and P.B. reconstituted F-actin pointed end samples for cryo-EM, and M.B.S., W.O. and O.H. collected the cryo-EM data. M.B.S. processed the cryo-EM data for the undecorated pointed end dataset and DNase I-bound dataset, while W.O. processed the cryo-EM data for the phalloidin-bound dataset. M.B.S. built all protein models and, together with W.O., analyzed the structures. M.B.S., W.O., P.B. and S.R. wrote the manuscript.

## Funding

## Competing interests
The authors declare no competing interests.
