## [Peer Review File · Nature Communications]

Phalloidin and DNase I-bound F-actin pointed end structures reveal principles of filament stabilization and disassemblyREVIEWER COMMENTS

Reviewer #1 (Remarks to the Author):

The authors reported the structure of the actin filament pointed end in the presence of phalloidin, as well as the structure stabilized by phalloidin and bound to DNase I. Unexpectedly, the pointed end subunits bound to phalloidin maintained an F-actin-like form (F-form), unlike the previously reported bare pointed end. This indicates how phalloidin inhibits depolymerization at the pointed end. The pointed end stabilized by phalloidin bound two DNase I molecules, and the structure suggested a mechanism by which DNase I depolymerizes the actin filament. Phalloidin is widely used in research on actin dynamics. Considering the importance of actin dynamics, I believe this manuscript is worth publishing with minor changes, although the biological significance of DNase I in actin dynamics might be relatively limited.

Minor Points:

1. The authors discussed that P0 and P1 subunits are in a G-actin-like form (G-form) without phalloidin and an F-form with phalloidin. However, this is not obvious from the current presentation (Fig 1d and supplementary Fig. 3). I believe views from the rotational axis between the domains with aligned inner domains would better illustrate the form change. Comparison of these subunits with previously reported typical F-form and G-form actins is also important.
2. Lines 179-181: The authors wrote that SD2s of actin P1 and P0 are superimposed. However, the main part of the D-loop was missing in the model without phalloidin. The authors should clarify that the alignment did not include the D-loop, as the D-loop is considered a major part of SD2.
3. The last part of the discussion: It is true that P1 and B-1 should perform the G to F transition for polymerization while P2 and B0 should perform the F to G transition for depolymerization. In the authors' model, additional potential barriers should be created by P0-P1, P2-P1, and P3-P1 interactions for the G to F transition and by P2-P4 and P2-P3 interactions for the F to G transition. The rate difference between the pointed end and the barbed end is substantially large. Are there any candidates for inducing the potential barrier?

Reviewer #2 (Remarks to the Author):

The manuscript by Sanders et al provides the structures of the pointed end of the actin filament in three states – undecorated, stabilized by phalloidin, and complexed with DNase-I and phalloidin. Technically, the work is done well and the manuscript is clearly written. Nevertheless, the first part of the paper, devoted to the structure of the undecorated pointed end, represents confirmatory studies that prove robustness of the previously published structures of the pointed end by Dominguez and Pollard labs. Importantly, Carman et al. 2023 resolved the structure of the pointed end to a higher (2.8 Å) resolution than reported in the manuscript (3.1 Å). The mechanism of the pointed end stabilization by phalloidin has novelty, but may be of interest to only a small audience.

The proposed model of actin depolymerization by the DNase-I is somewhat speculative since it is based on the assumption that DNase-I binds to the D-loop of the terminal phalloidin-free pointed end similarly to that observed when the F-actin's pointed end is stabilized with phalloidin. Overall, I think that this work is more appropriate for a more specialized journal.

Reviewer #3 (Remarks to the Author):

The assembly and disassembly of actin filaments are associated with conformational changes of actin subunits, and these processes can be further regulated by a number of actin-binding proteins and small molecule inhibitors, such as phalloidin. Here, the authors applied cryo-EM to elucidate the mechanisms by which phalloidin stabilizes actin filament pointed ends and how DNaseI accelerates filament pointed end depolymerization. They confirm recent findings that the two terminal subunits at the pointed end adopt a G-actin -like conformation, and reveal that phalloidin induces a flattened, F-actin -like conformation of these two terminal subunits by directly bridging them. They also show that DNaseI can associate with the two terminal subunits of phalloidin-stabilized filament pointed end, and propose a mechanism by which DNaseI accelerates filament depolymerization in the absence of phalloidin through a steric clash between the ultimate actin subunit and the DNaseI molecule bound to the penultimate actin subunit. Finally, the authors present an interesting model, that is based on G-to-F and F-to-G transitions, which may explain the kinetic asymmetry of actin filament barbed and pointed ends.

The structural work presented in the manuscript appears of very good quality, and the study provides valuable new information of the regulation of actin filament pointed end dynamics. However, there are few relatively small points that should be addressed to further strengthen this study.

1. It is interesting that phalloidin binds also between the two last actin subunits (P0 and P1) at the filament pointed end. However, this binding site is not 'complete', and thus one would assume that the affinity of phalloidin to the filament pointed end may be lower as compared to its binding sites along the filament. Is there any evidence indicating that a higher concentration of phalloidin would be needed to stabilize filament pointed ends as compared to barbed ends? This could be tested, or at least discussed in the manuscript.

2. The model presented in figure 6 is potentially very important. This model also predicts that a small fraction of P1 (or P2) subunits would spontaneously undergo G-to-F or F-to-G transition. Did the authors find any support for the presence of such transitions in their structures (i.e. were there any evidence of structural heterogeneity of these subunits in the cryo-EM maps)?

3. Based on the 'Methods' and Tables 1 and 2, different actin isoforms were used in the structures presented in the manuscript. This should be mentioned also in the 'Results' for clarity.

4. Legends to Fig. 5 and Fig. 6 need more information/explanation to make them better accessible to a non-specialist reader. Furthermore, the authors should specify in the legend for Fig. 1 what the arrows indicate in panel d.

Point-to-point response to the reviewers' comments

We gratefully thank the reviewers for their positive feedback and insightful comments, which aided us to further improve the manuscript. Below is a point-by-point response to all comments and a detailed description of all changes we have made to our manuscript after considering their suggestions. The changes are highlighted in yellow in the revised text.

Reviewer #1 (Remarks to the Author):

The authors reported the structure of the actin filament pointed end in the presence of phalloidin, as well as the structure stabilized by phalloidin and bound to DNase I. Unexpectedly, the pointed end subunits bound to phalloidin maintained an F-actin-like form (F-form), unlike the previously reported bare pointed end. This indicates how phalloidin inhibits depolymerization at the pointed end. The pointed end stabilized by phalloidin bound two DNase I molecules, and the structure suggested a mechanism by which DNase I depolymerizes the actin filament. Phalloidin is widely used in research on actin dynamics. Considering the importance of actin dynamics, I believe this manuscript is worth publishing with minor changes, although the biological significance of DNase I in actin dynamics might be relatively limited.

Reply 1.0: We thank the reviewer for their positive feedback. While we agree that the role of DNase I in controlling actin dynamics may be limited, the interaction of DNase I with actin in general is of great biological importance. DNase I facilitates chromatin breakdown during apoptosis and cells secrete it to catalyze the hydrolysis of extracellular DNA. However, it is imperative that DNase I present in the healthy cell remains inactive to not degrade the cell's own DNA. And this is precisely where the interaction with actin is highly relevant. The actin binding and DNA cleaving activities of DNase I are mutually exclusive. Hence, actin is a potent inhibitor of the DNA cleavage activity of DNase I, especially because it is present at high concentration inside cells, as a monomer or filament, and is actively removed from the blood stream by the actin scavenger system (PMID: 389627, 477868, 2154744). Inside cells, the filament depolymerization activity of DNase I ensures that the enzyme is inactivated even in microenvironments where the monomer concentration might be low. Therefore, we believe that understanding the mechanism of actin filament depolymerization by DNase I is relevant to a large audience.

Minor Points:

[1.1] The authors discussed that P0 and P1 subunits are in a G-actin-like form (G-form) without phalloidin and an F-form with phalloidin. However, this is not obvious from the current presentation (Fig 1d and supplementary Fig. 3). I believe views from the rotational axis between the domains with aligned inner domains would better illustrate the form change. Comparison of these subunits with previously reported typical F-form and G-form actins is also important.

Reply 1.1: We thank the reviewer for their helpful input. The actin molecules in figure 1c and supplementary figure 3 are now shown along the rotation axis between the domains, according to the recommendation. The comparisons of these subunits with previously reported G-actin (pdb 3HBT) and F-actin (pdb 8A2T) were already present in supplementary figure 3.

[1.2] Lines 179-181: The authors wrote that SD2s of actin P1 and P0 are superimposed. However, the main part of the D-loop was missing in the model without phalloidin. The authors should clarify that the alignment did not include the D-loop, as the D-loop is considered a major part of SD2.

Reply 1.2: That is indeed the case, as the command used to superimpose the structures (matchmaker) first performs a pairwise sequence alignment and then fits only on the aligned residues. The following sentence has been added in line 183: “The alignment was performed by superimposing the resolved region of SD2 for both structures (aa 34-39,50-69).”

[1.3] The last part of the discussion: It is true that P1 and B-1 should perform the G to F transition for polymerization while P2 and B0 should perform the F to G transition for depolymerization. In the authors’ model, additional potential barriers should be created by P0-P1, P2-P1, and P3-P1 interactions for the G to F transition and by P2-P4 and P2-P3 interactions for the F to G transition. The rate difference between the pointed end and the barbed end is substantially large. Are there any candidates for inducing the potential barrier?

Reply 1.3: This is a very interesting question, which motivated us to inspect our structures more closely. By comparing the undecorated pointed end structure with the phalloidin-bound pointed end structure, where the last two subunits remain in the F-form, we were indeed able to propose some candidate residues.

For the G-to-F transition, the most notable interactions that need to rearrange during flattening can be found in the interface between subunits P₁-P₃, more specifically between P₁ SD1 and the D-loop of P₃. As prominent examples, amino acids P₁-Y133 and P₁-I136 form interactions with the D-loop that are disrupted during flattening, which could pose a kinetic barrier. Accordingly, these interactions are irrelevant for the subunit that needs to flatten at the barbed end (B-1).

For to F-to-G transition, the most drastic change we observe is the rupture of the salt bridge between P₁-K113 and P₂-E195. This interaction has been proposed to influence actin depolymerization, since it acts as a catch-slip bond under force (PMID: 23460697). In addition, the salt bridge between P₀-E270 and P₁-R39 needs to rearrange to form a new salt bridge between P₀-E270 and P₁-R62. At the interaction site with the D-loop of the lower subunit (P₄), some hydrophobic residues move away from the hydrophobic pocket, with the most prominent examples being P₁-L349 and the C-terminal P₁-F375.

We added a short statement in the discussion and a new supplementary figure to illustrate these rearrangements (Supplementary figure 9). While we believe that it would be interesting to further investigate the potential barrier by exploring the effect of mutations on actin-end dynamics, this is not within the scope of our current study.

Reviewer #2 (Remarks to the Author):

[2.1] The manuscript by Sanders et al provides the structures of the pointed end of the actin filament in three states – undecorated, stabilized by phalloidin, and complexed with DNase-I and phalloidin. Technically, the work is done well and the manuscript is clearly written.

Reply 2.1: We thank the reviewer for their positive feedback on the quality of the work and manuscript.

[2.2] Nevertheless, the first part of the paper, devoted to the structure of the undecorated pointed end, represents confirmatory studies that prove robustness of the previously published structures of the pointed end by Dominguez and Pollard labs.

Reply 2.2: We believe that confirming or refuting prior work is an important part of scientific progress. The relevant published article is cited and acknowledged in our manuscript and we never claimed novelty on the structure of the undecorated pointed end. Most importantly, we respectfully disagree with the reviewer that the first part of the paper is ‘devoted to the structure of the undecorated pointed end’. Instead, the structure of the undecorated pointed end is mentioned only briefly and mainly serves as a reference to enable the direct comparison with the novel structure of the phalloidin-stabilized pointed end, which constitutes the main focus of the first half of the manuscript.

[2.3] Importantly, Carman et al. 2023 resolved the structure of the pointed end to a higher (2.8 Å) resolution than that reported in the manuscript (3.1 Å).

Reply 2.3: While it is a useful metric in cryo-EM, we believe that the resolution (especially as a single number) should not be used to compare cryo-EM density maps directly. Instead, one should inspect the maps individually to assess the quality of resolved domains, density for individual sidechains etc. Based on inspection of the density maps, we would argue that the quality of both is comparable. For both maps, the resolution is high enough to distinguish side chains for most of the pointed end.

[2.4] The mechanism of the pointed end stabilization by phalloidin has novelty, but may be of interest to only a small audience.

Reply 2.4: Phalloidin is a widely used molecular tool for studying the actin cytoskeleton. It is a robust fluorescence probe for visualizing F-actin structures in cells using fluorescence microscopy; and it is also commonly employed to study the effects of arresting actin dynamics, in combination with monomer sequestering drugs such as latrunculin. Thus, our data on phalloidin are highly relevant for the many scientists working in the cytoskeleton and muscle fields. This is also acknowledged by the other two reviewers. To give a specific example: we show in Supplementary Fig. 8 that the binding site of phalloidin overlaps with that of tropomodulin. Our data therefore indicate that phalloidin should be used with great caution when staining actin filaments in muscle sarcomeres, because its addition may lead to the disruption of the pointed end arrangement. Hence, we respectfully disagree with the reviewer that our structures are only relevant to a small audience.

[2.5] The proposed model of actin depolymerization by the DNase-I is somewhat speculative since it is based on the assumption that DNase-I binds to the D-loop of the terminal phalloidin-free pointed end similarly to that observed when the F-actin’s pointed end is stabilized with phalloidin.

Reply 2.5: While the reviewer is correct that we have not directly visualized the binding of DNase I to the D-loop of actin subunits at the pointed end in the absence of phalloidin, we believe that the

proposed binding mode is warranted. The binding of DNase I to the D-loop is highly specific, because DNase I forms a tight β -sheet-like interaction with the D-loop at nanomolar affinity. Accordingly, the observed interface between actin and DNase I is essentially the same when comparing crystal structures of G-actin bound by DNase I and our phalloidin-stabilized pointed end bound by DNase I. In fact, in the phalloidin-stabilized structures, we see that SD2 of actin subunits at the pointed end is very flexible, resulting in the ‘wobbling’ of DNase I molecules (Fig. 3c). Despite this extensive flexibility, the actin-DNase I interface remains the same. Moreover, because the affinities of DNase I for G-actin and F-actin are comparable, we do not expect a different binding mode in the absence of phalloidin. This is consistent with a wealth of biochemical data that have revealed that DNase I binds to the pointed end of the filament and not to the filament center or barbed end.

Reviewer #3 (Remarks to the Author):

The assembly and disassembly of actin filaments are associated with conformational changes of actin subunits, and these processes can be further regulated by a number of actin-binding proteins and small molecule inhibitors, such as phalloidin. Here, the authors applied cryo-EM to elucidate the mechanisms by which phalloidin stabilizes actin filament pointed ends and how DNaseI accelerates filament pointed end depolymerization. They confirm recent findings that the two terminal subunits at the pointed end adopt a G-actin -like conformation, and reveal that phalloidin induces a flattened, F-actin -like conformation of these two terminal subunits by directly bridging them. They also show that DNaseI can associate with the two terminal subunits of phalloidin-stabilized filament pointed end, and propose a mechanism by which DNaseI accelerates filament depolymerization in the absence of phalloidin through a steric clash between the ultimate actin subunit and the DNaseI molecule bound to the penultimate actin subunit. Finally, the authors present an interesting model, that is based on G-to-F and F-to-G transitions, which may explain the kinetic asymmetry of actin filament barbed and pointed ends.

The structural work presented in the manuscript appears of very good quality, and the study provides valuable new information of the regulation of actin filament pointed end dynamics. However, there are few relatively small points that should be addressed to further strengthen this study.

Reply 3.0: We thank the reviewer for their positive feedback on our work.

[3.1] It is interesting that phalloidin binds also between the two last actin subunits (P0 and P1) at the filament pointed end. However, this binding site is not ‘complete’, and thus one would assume that the affinity of phalloidin to the filament pointed end may be lower as compared to its binding sites along the filament. Is there any evidence indicating that a higher concentration of phalloidin would be needed to stabilize filament pointed ends as compared to barbed ends? This could be tested, or at least discussed in the manuscript.

Reply 3.1: This is a very interesting question, which we now address in a short paragraph in the discussion section of the manuscript.

At the pointed end, we observe a phalloidin molecule bridging subunits P₀, P₁ and P₂, but also an occupied ‘incomplete’ binding site between just P₀ and P₁. At the barbed end, this is different. In our previously determined structure of the phalloidin-stabilized barbed end (see pdb 8OI6, PMID: 37749275), the ultimate phalloidin molecule bridges subunits B₀, B₁ and B₂ (Reviewer Fig. 1). Notably, in this structure, we did not observe density for phalloidin at the potential ‘incomplete’ binding site at the barbed end situated between B₀ and B₁ (Reviewer Fig. 1). This indicates that phalloidin has a stronger stabilizing function at the pointed end compared to the barbed end, because the binding of the “incomplete binding site” introduces additional bridges.

The reviewer is correct in assuming that the binding affinity of phalloidin for this incomplete binding site might be lower compared to all other (complete) binding sites. However, even if this terminal phalloidin would not be present, the ultimate subunit would still bound by a second phalloidin molecule at its base. It is therefore difficult to correlate phalloidin binding to the ‘incomplete’ site directly with pointed end stability. We are unaware of previous measurements in literature to address the type of asymmetry hypothesized by the reviewer. Early in the project, we contemplated performing an experiment to directly visualize phalloidin turnover at the filament end via single molecule TIRF microscopy. However, the large number of internal binding site combined with the slow binding/unbinding kinetics of phalloidin to the filament (PMID: 7981198) would make such an experiment technically extremely challenging and very difficult to interpret.

Reviewer Fig. 1. Cryo-EM density map of the barbed end (PMID: 37749275, EMD-16887), shown in two orientations. The ultimate actin subunit is (B₀) colored orange, the penultimate actin subunit (B₁) is colored salmon, and subunits towards the filament center (B₂ and above) are colored dark red. Phalloidin is shown in cyan and ADP in yellow.

[3.2] The model presented in figure 6 is potentially very important. This model also predicts that a small fraction of P₁ (or P₂) subunits would spontaneously undergo G-to-F or F-to-G transition. Did the authors find any support for the presence of such transitions in their structures (i.e. were there any evidence of structural heterogeneity of these subunits in the cryo-EM maps)?

Reply 3.2: We did not find any evidence of heterogeneity in our structures, and the pointed end structure reached high resolution without the need of extensive classification of particles. However, this does not exclude the possibility of a very small fraction of subunits spontaneously

changing their conformation from G-to-F or vice versa; if it is a very short lived state we would not capture it in the cryo-EM data. Our phalloidin-bound pointed end structure suggests that these subunits can potentially undergo this conformational change, although it is also possible that a trigger is needed for this to happen (for example, docking of the D-loop or phalloidin binding).

[3.3] Based on the ‘Methods’ and Tables 1 and 2, different actin isoforms were used in the structures presented in the manuscript. This should be mentioned also in the ‘Results’ for clarity.

Reply 3.3: We thank the reviewer for pointing this out. The actin isoforms used are now specified at the beginning of each results section.

[3.4] Legends to Fig. 5 and Fig. 6 need more information/explanation to make them better accessible to a non-specialist reader. Furthermore, the authors should specify in the legend for Fig. 1 what the arrows indicate in panel d.

Reply 3.4: This is a great suggestion. We added legends for Fig. 5 and Fig. 6, as well as specifying the meaning of the arrow in Fig. 1.

REVIEWERS' COMMENTS

Reviewer #1 (Remarks to the Author):

The authors have satisfactorily addressed my comments. I have no further remarks and believe the current manuscript is suitable for publication.

Reviewer #2 (Remarks to the Author):

I commend the authors for their detailed response to all the points raised by the reviewers.

Reviewer #3 (Remarks to the Author):

The authors have satisfactorily addressed my previous concerns/comments.